# Neuroactive Amino Acid Profile in Autism Spectrum Disorder: Results from a Clinical Sample

**DOI:** 10.3390/children10020412

**Published:** 2023-02-20

**Authors:** Martina Randazzo, Adriana Prato, MariaAnna Messina, Concetta Meli, Antonino Casabona, Renata Rizzo, Rita Barone

**Affiliations:** 1Child Neurology and Psychiatry, Department of Clinical and Experimental Medicine, University of Catania, 95123 Catania, Italy; 2Referral Centre for Inherited Metabolic Disorders, Department of Clinical and Experimental Medicine, University of Catania, 95123 Catania, Italy; 3Department of Biomedical and Biotechnological Sciences, Section of Physiology, University of Catania, 95123 Catania, Italy; 4Reseach Unit of Rare Diseases and Neurodevelopmental Disorders, Oasi Research Institute-IRCCS, 94018 Troina, Italy

**Keywords:** autism spectrum disorder, dried blood spots, ESI-MS/MS, metabolism, amino acids, intellectual disability

## Abstract

Biological bases of autism spectrum disorder (ASD) include both genetic and epigenetic causes. Patients with ASD show anomalies in the profile of certain plasma amino acids, including neuroactive amino acids. Monitoring plasma amino acids may be relevant for patient care and interventions. We evaluated the plasma amino acid profile in samples extracted from dry blood spots by electrospray ionization-tandem mass spectrometry. Fourteen amino acids and eleven amino acid ratios were examined in patients with ASD and intellectual disability (ID), and neurotypical control subjects (TD). The amino acid profile in the ASD group showed reduced levels of ornithine (*p* = 0.008), phenylalanine (*p* = 0.042) and tyrosine (*p* = 0.013). The statistically significant amino acid ratios were Leu+Val/Phe+Tyr (*p* = 0.002), Tyr/Leu (*p* = 0.007) and Val/Phe (*p* = 0.028), such differences remaining significant only in the comparison between ASD and TD. Finally, a positive correlation emerged between the score of the restricted and repetitive behavior on ADOS-2 and the citrulline levels in the ASD group (*p* = 0.0047). To conclude, patients with ASD may show a distinguishable metabolic profile useful for studying their metabolic pathways in order to develop screening tests and targeted therapies.

## 1. Introduction

Autism spectrum disorder (ASD) is a neurodevelopmental disorder characterized by deficits in social communication and social interaction, along with restricted interests and repetitive behaviors. Several specifiers are used to outline the severity of key symptoms that accompany ASD definition, inter alia, verbal language and cognitive levels [1]. Furthermore, 45% of ASD patients have comorbidities such as intellectual disability, while almost 30% have a regression or loss of gained skills [2]. Co-occurring disorders, including attention deficit/hyperactivity disorder, anxiety disorders, oppositional defiant disorder, chronic tic disorder and obsessive–compulsive disorder, contribute to ASD morbidity [2]. ASD is a neurobiological multifactorial disorder attributable to both genetic and environmental factors [2]. Amino acids (AA) are of crucial importance in brain function, not only as metabolic intermediates and building blocks of proteins, but also as mediators of inter-neuronal communication [3]. Alterations in peripheral AA may be correlated with central brain functions, and changes in amino acid availability and/or metabolism may contribute to the pathogenesis of psychiatric disorders. Body fluid levels of neuroactive AA, including glutamate, glutamine, taurine, gamma-aminobutyric acid (GABA), glycine, tryptophan, serine and others have been widely studied in ASD, but the results reported in the literature are variable [3]. In ASD patients, malabsorption and food selectivity may cause changes in amino acid concentrations; therefore, monitoring blood amino acid levels may be helpful in the management of people with ASD [4]. Neuroactive AA changes have been described in psychiatric disorders that have symptoms in common with ASD, such as cognitive impairment and issues with social interactions [5]. As a consequence of the potential role of neuroactive AA, monitoring alterations in their concentrations in body fluids is crucial in case they are relevant to the early diagnosis of and targeted intervention in patients with ASD. The aim of the present study is to recognize the occurrence of neuroactive AA changes in ASD patients for a finer understanding of molecular mechanisms that can lead to customized screenings and therapies.

## 2. Materials and Methods

The present cross-sectional study included 89 participants recruited at the Child Neurology and Psychiatry Unit, Department of Clinical and Experimental Medicine, University of Catania, in a two-year period, between June 2019 and June 2021. Participants included 40 individuals with ASD, 22 with intellectual disability (ID) and 27 with typical development (TD) (Figure 1). Patients (n: 62) were recruited among a clinical population of children and adolescents with neurodevelopmental disturbances admitted at the outpatient clinic in the study period. We ascertained patients, without restrictions of gender or ethnicity, who referred to our department because of neurological, behavioral and developmental disorders that appeared to be related to ASD or ID.

The diagnosis of ASD was obtained according to DSM-5 criteria [1] and using gold-standard standardized diagnostic tests including the Autism Diagnostic Interview—Revised (ADI-R) and Autism Diagnostic Observation Schedule (ADOS). ADI-R is an investigator-based parent or caregiver interview that yielded a description of history, as well as current functioning, in areas of development related to autism [6]. ADOS is a semistructured, standardized assessment of social affect (SA), which comprehends language, communication and social reciprocal interaction, and restricted and repetitive behaviors (RRB) for individuals suspected of having ASD [7]. No child had any diagnosed genetic, metabolic or neurological etiology for ASD. Children with secondary autism such as those with tuberous sclerosis or Fragile X syndrome or causative copy number variants (CNV) were excluded from the study (n: 47). ASD patients with extreme food selectivity were excluded as well to rule out interferences of restricted diets on blood AA levels (n: 13).

The standardized diagnosis of ID was based on DSM-5 criteria. Developmental quotient (DQ) and/or Intellectual quotient (IQ) were measured in all participants by a comprehensive, standardized neuropsychological assessment battery administered according to age [8,9]. Children with secondary or syndromic causes of ID were excluded from the study (n: 21). As aforementioned, ID subjects with extreme food selectivity were not included to avoid the possible impact of restricted diets on blood AA levels (n: 7).

As a control group (n: 27), we included subjects from a community sample with no neurodevelopmental disturbances and with an age and gender distribution equal to the patients with ASD or ID. Neurotypical children with food selectivity or special dietary habits were not included (n: 13). All the participants were screened by means of anamnestic interviews with their parents for present or past medical conditions and pharmacological therapy. Clinical features, metabolic assessment, cognitive levels and measure of severity of autism symptoms were analyzed. To avoid any systematic difference related to the time of sample collection, all patients underwent blood collection in the morning between 8.00 and 8.30 a.m. after nocturnal fasting. Routine blood analyses included relevant analytes for possible metabolic disarrangement such as basal serum glucose, glutamate-pyruvate transaminase, creatine kinase, vitamin D, lactate and ammonia levels. For the same reason mentioned above, blood spots on filter paper card (Whatman card Specimen 903) were collected from each participant in the ASD, ID and TD groups in the morning between 8.00 and 8.30 a.m. after nocturnal fasting. Once dried, blood spots were stored at 4° C in a unique refrigerator with controlled humidity rate and processed within 2 weeks of sampling. A 3.2 mm diameter blood dot of each individual was used for the analyses. Underivatized specimens were analyzed using electrospray ionization-tandem mass spectrometry (ESI-MS/MS). Twenty-five metabolites, including fourteen AA and eleven AA ratios, were simultaneously measured in Dried Blood Spot (DBS). The following AA were determined: alanine (Ala), arginine (Arg), citrulline (Cit), glutamate (Glu), glutamine (Gln), glycine (Gly), leucine (Leu), methionine (Met), ornithine (Orn), phenylalanine (Phe), proline (Pro), tyrosine (Tyr) and valine (Val). The following AA ratios were measured: Leu+Val/Phe+Tyr, Cit/Arg, Cit/Phe, Leu/Phe, Met/Leu, Met/Phe, Phe/Tyr, Glu/Gln, Tyr/Leu, Tyr/Met and Val/Phe. The analyte concentration was quantified by comparison with a known concentration of corresponding stable-isotope internal standards. All the procedures adopted for the current research were performed as part of the routine clinical assessment. All the results were collected in a database in order to classify the patients based on main clinical features and diagnosis of ASD, ID or TD (controls). Qualitative data were presented as a number and percentage. Quantitative data were expressed as mean and standard deviation. ANOVA (f) test was used for comparison between three groups with quantitative variables that distributed normally, while the effect size was estimated by measuring the partial eta squared (η^2^_p_) as previously described [10]. Furthermore, when the ANOVA was significant, post hoc pairwise comparisons were conducted and adjusted by Bonferroni correction. Possible associations between the identified metabolites and clinical features of ASD patients, including ASD symptom severity (ADOS 2 scores) and DQ/IQ scores, were verified by Spearman correlation analyses. For discrete or continuous data, the correlations were performed using the Pearson coefficient (r). Data were analyzed using IBM SPSS, version 25 (SPS S.r.l., Bologna, Italy). A comparison was considered significant if the corrected *p*-value was less than 0.05.

### Ethical Statement

This study was based on data collected as part of the clinical care of patients with neurodevelopmental disorders. All procedures follow recommendations of the local Ethical Committee and are in accordance with the 1964 Helsinki declaration and its later amendments. Written informed consent was signed by the parents of all participants. The study protocol was approved as part of a larger clinical study on autism spectrum disorder. The larger study was approved with n° 759 by the local Ethical committee at Policlinico University of Catania.

## 3. Results

The ASD group included 40 subjects (males: 32; females: 8; age: 6.98 ± 4.03 years) and the ID group included 22 subjects (males: 13; females: 9; age: 7.18 ± 2.97 years). A total of 27 TD subjects (males: 22; females: 5; age: 5.96 ± 4.22) was used as a control group. ANOVA analysis showed no significant discrepancy in age among the three groups (F = 0.764, *p* = 0.469).

Mean developmental/intellectual quotient (DQ/IQ) measured in the ASD group was 64.94 ± 17.08, and in the ID group was 59.67 ± 14.54 (normal average IQ range: 90–109). Furthermore, in standardized diagnostic test ADOS, the ASD group showed mean SA score of 14.24 ± 4.29 and mean RRB score of 3.48 ± 1.75, indicating moderate to severe autism symptom severity (DSM-5 severity levels 2/3). 

As for routine laboratory blood analyses, we found no significant differences among the study groups when comparing hemoglobin (F = 0.134; *p* = 0.875), glycemia (F = 0.573; *p* = 0.567), glutamate-pyruvate transaminase (F = 0.375; *p* = 0.689), creatine phosphokinase (F = 0.099; *p* = 0.906) and vitamin D (F = 0.727; *p* = 0.490). Notably, serum lactate was significantly higher in ASD (F = 3.2; *p* = 0.047). Lactate levels tended to be higher in the group with ASD (20.30 ±15.61) compared with the ID group (10.69 ± 4.41) (*p* = 0.052). 

The amino acid profile in patients with ASD showed statistically significant decreased ornithine (*p* = 0.008, η^2^_p_ = 0.11), phenylalanine (*p* = 0.042, η^2^_p_ = 0.07) and tyrosine blood levels (*p* = 0.013, η^2^_p_ = 0.1) (Table 1). We did not find any other significant differences in the remaining analyzed AA among study groups.

Pairwise comparison with Bonferroni correction showed that ornithine (*p* = 0.006, Cohen’s d = 0.70), phenylalanine (*p* = 0.043, Cohen’s d = 0.59) and tyrosine (*p* = 0.010, Cohen’s d = 0.71) blood levels were significantly lower in the ASD group compared with the TD group (Table 1).

We further analyzed several ratios between the studied AA: the Leu+Val/Phe+Tyr ratio was significantly different among groups in the ANOVA test (*p* = 0.002, η^2^_p_ = 0.13). Pairwise comparison demonstrated that a significant increase occurred when comparing ASD versus TD (*p* = 0.003). The Tyr/Leu ratio difference was significant in ANOVA testing (*p* = 0.007, η^2^_p_ = 0.11), and its levels appeared reduced in the ASD group in comparison with both the TD (*p* = 0.044) and ID group (*p* = 0.017). Finally, ANOVA showed a significant difference in Val/Phe among the three groups (*p* = 0.028, η2p = 0.08), and its levels in the ASD group compared with TD appeared significantly higher (*p* = 0.023). No other ratios were statistically different among the study groups. In sum, the differences we ascertained may be ascribed mainly to variations in the amino acid profile in patients with ASD (Table 2).

Finally, we evaluated the number of ASD subjects (%) presenting with individual amino acid or ratio outside the reference range (RR) (Table 3). For this purpose, we established normal reference ranges in the TD group and assessed the tenth and ninetieth percentiles of their distribution. For the majority of metabolites, most ASD patients fell within the normal range. Nonetheless, for metabolites significantly decreased in the ASD group, we found that almost 15–22.5% of subjects with ASD fell below the lowest normal values. Conversely, for the same metabolites, a very limited number (Phe 2.5%) or none of the ASD subjects fell above the RR (e.g., Orn, Tyr and Tyr/Leu 0%). Likewise, for amino acid ratios significantly increased in the ASD group, almost 30–35% of patients were above the upper normal limit. Notably, for these ratios, none or a small percentage of ASD subjects lied below the RR (Leu+Val/Phe+Tyr 0%, Val/Phe 5%).

We conducted a Spearman correlation test to find out possible correlations among the aforementioned metabolites and clinical features of ASD patients, e.g., intelligence quotient (IQ) or developmental quotient (DQ), ADOS-2 SA, ADOS-2 RRB and ADOS-2 Total Score (ADOS-2 TS). We found a statistically significant positive correlation between RRB score and the citrulline levels in patients with ASD (r = 0.4801; *p* = 0.0047) (Table 4).

## 4. Discussion

ASD is a polygenic multifactorial disorder, and in recent years, there has been a growing interest in the potential role of neuroactive AA in the pathogenesis, early diagnosis and treatment of ASD. In the present study, of the twenty-five metabolites (fourteen AA and eleven AA ratios) analyzed in the three groups, six (24%) proved statistically significant differences: ornithine, phenylalanine, tyrosine, Leu+Val/Phe+Tyr, Tyr/Leu and Val/Phe. The biological bases of ASD are extremely heterogeneous, and several lines of evidence suggest searching for the distribution of observed metabolic abnormalities in studied clinical samples [11]. We found that almost 20% of patients with ASD had aromatic amino acid levels (Phe, Tyr) far below the normal low limit. To confirm these data and to better understand the metabolic interplay among measured metabolites, we evaluated pertinent amino acid ratios. We found that almost 30–35% of subjects in the ASD group had increased ratios of branched-chain amino acids (BCAAs) such as leucine and valine with aromatic amino acids (Leu+Val/Phe+Tyr; Val/Phe). As a whole, the observed significant blood amino acid changes suggest defect in aromatic AA biosynthetic pathways, which are related to neurotransmitters and neuromodulators synthesis, thyroid hormones and protein biosynthesis. In particular, abnormal phenylalanine in ASD are consistent with the results in Arnold et al. [12], Adams et al. [13], Tirouvanziam et al. [14] and El Baz et al. [15]. Furthermore, Tyrosine and tryptophan metabolism have been repeatedly associated with ASD [16,17,18,19,20] and recently related to increased aberrant disruptive behavior in children with ASD without neurodevelopmental regression [21]. In a population of 403 children, plasma and stool metabolomics analyses at age 3 years showed convergent results, indicating the disruption of tyrosine metabolism, which was related with autism-risk poor communication scores measured on the Ages and Stages Questionnaire (ASQ) [22]. Large neutral amino acids, including both aromatic and BCAA, are transported across the blood–brain barrier (BBB) by means of the large amino acid transporter (LAT) protein complexes 1 and 2, which are expressed in both blood and brain. Genetic variants of LAT subunits have been associated with impaired amino acid transport in the BBB in Slc7a5^−/−^ mice with an ASD phenotype [23]. Moreover, 17% of studied patients with ASD had LAT coding variants and exhibited abnormal utilization of large neutral amino acids [24]. Altogether, it could be argued that the transport of neutral AA through the brain capillary endothelial wall, which makes up the BBB in vivo, is an important check point for the overall regulation of cerebral metabolism, including protein synthesis and neurotransmitter production; a neutral amino acid transporter related to ASD regulates the flux of several neutral AA implicated in the present analyses.

An increasing number of studies on neuroactive amino acids in ASD focused on glutamic acid metabolism, since glutamate is the major excitatory neurotransmitter and is implicated in idiopathic autism as well as in syndromic autism conditions such as Fragile-X syndrome [25,26]. A comprehensive meta-analysis pooling twelve studies for a total of 880 participants considered the effect size and between-study heterogeneity for the association between blood glutamate levels and ASD. Blood glutamate levels were increased in ASD [Standardized Mean Difference, SMD = 0.99, 95% confidence interval (95% CI) = 0.58–1.40; *p* < 0.001)] compared with controls, although a high heterogeneity across studies was observed (I = 86%, *p* < 0.001) [27]. We found that, on average, glutamate levels in blood were in the normal ranges in the study groups and a minority of ASD samples (7.5%) had glutamate levels above the upper normal levels. Some issues related to biological samples used for the analyses, participants characteristics, dietary habits and concurrent therapy may add to discrepancies among studies [27], including the present one. Several studies afforded amino acid metabolism in relation with oxidative stress in ASD. A pooled meta-analysis on oxidative stress biomarkers in ASD including 87 studies found abnormal values of several metabolites involved in the trans-methylation cycle and trans-sulfuration pathways. Among other metabolites, methionine was significantly decreased (*p* < 0.001), while glutathione metabolism markers and vitamin D showed the larger effect sizes for the association with ASD [28]. Methionine is synthesized using cysteine in the trans-sulfuration pathway, which connects methionine and glutathione biosynthesis. Interestingly, methionine levels were below the normal range in almost 22% of the ASD group in our cohort, thus supporting a decreased antioxidant reservoir in ASD [29].

Measuring amino acid concentrations provides a valuable indicator of metabolic activity associated with many tissues in the body [30]. In ASD patients, malabsorption and protein metabolism might also cause changes in amino acid concentrations. Thus, the measurement of exogenous amino acids can be useful in the management of ASD patients with dietary issues [4]. To our knowledge, decreased values of ornithine are an isolated outcome in this field of research. In a large study of the metabolism of children with ASD (CAMP study), it was shown that ratios of some metabolites could identify subpopulations that exhibit the dysregulation of AA metabolism associated with ASD [30]. In particular, using ratios of glutamine, glycine, ornithine and serine with leucine, isoleucine and valine performed efficiently to identify participants with ASD in 17% of CAMP ASD subjects with a specificity of 96.3% and PPV of 93.5% [31]. All statistically significant ratios in our study, interestingly, reflect the balance between branched-chain amino acids, such as leucine, isoleucine and valine, and aromatic amino acids, e.g., phenylalanine and tyrosine. Mutations in amino acid transporters and enzymes involved in the metabolism of branched-chain amino acids (BCAAs) are relevant in ASD. Remarkably, Novarino et al. (2012) described a mendelian form of autism with comorbid ID and epilepsy associated with low plasma BCAAs levels that normalized after dietary supplementation [32]. The branched-chain α-keto acid dehydrogenase kinase (bckdk) knockout mice showed decreased BCAA brain levels and displayed neurobehavioral defects reversed by BCAAs supplementation [32]. Later on, two unrelated children were found to have two new mutations in the BCKDK gene, and BCAA levels in their body fluids were decreased. Their developmental delay and abnormal neurobehaviors were also partially corrected after BCAA supplementation with a protein-rich diet [33,34].

In the present study, we found a positive correlation which resulted statistically significant between citrulline levels and stereotyped behavior (RRB score on ADOS-2) in children with ASD. In a previous study, we described significantly augmented citrulline levels in children with ASD [35]. Interestingly, Waisbren et al. [36] proved that cumulative exposure to citrulline and ammonia are the most reliable markers of poorer cognitive functioning in patients with classic citrullinemia (argininosuccinic acid synthetase deficiency). They showed that increased mean lifetime and cumulative citrulline levels were highly associated with intellectual functioning, verbal skills and visual performance. No significant correlations were found among studied clinical parameters and amino acid changes in the study samples.

## 5. Conclusions

The relationship between metabolic profile disturbances and neurodevelopmental disorders is still in its infancy. The present study concurs with previous research in showing that ASD subtypes may be associated with metabolomics changes focusing on the aromatic amino acids and BCAAs metabolism. Study limitations are represented by the small sample size and limited number of measured metabolites. At the same time, it is important to take into account the intrinsic heterogeneity of symptoms and severity in ASD. In particular, the large majority of studied patients with ASD were cognitively impaired and had moderate to severe autism symptom severity. Such factors prevented us from comparing the amino acid profiles between ASD subjects with different severity levels, according to the DSM-5 definition (level 1 versus levels 2–3). Moreover, the recruited participants were rather homogeneous in ages, precluding us to compare younger and older participants with regard to blood amino acid variations. On the other hand, we used a high-throughput analysis and systematically compared ASD with ID/DD and TD subjects, thus supporting meaningful amino acid changes in the ASD group. Though there exist controversial results, present findings and those in the literature support amino acid profile analyses in ASD patients to better understand those metabolic pathways amenable by therapeutic interventions.

## Figures and Tables

**Figure 1 children-10-00412-f001:**
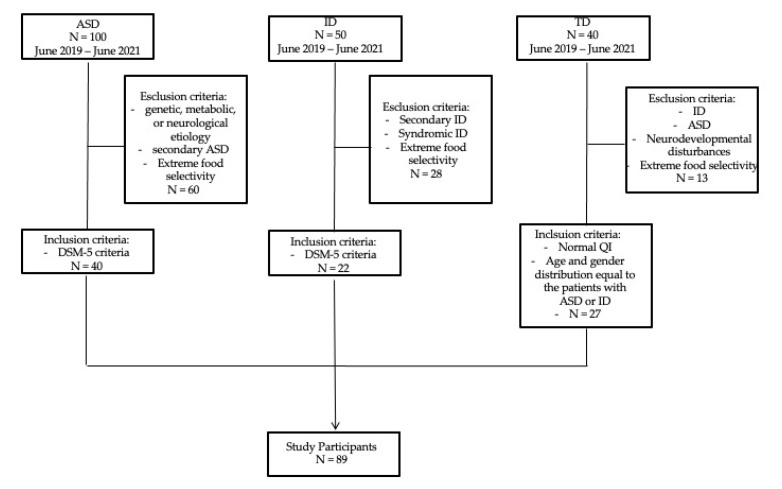
Recruitment Flow Diagram. ASD = Autism Spectrum Disorder; ID = Intellectual Disability; TD = Typical Development.

**Table 1 children-10-00412-t001:** Amino acid levels comparison with ANOVA test in the three studied groups and paired post hoc comparisons with Bonferroni pairwise correction.

Amino Acid	ASD (μmol/L)	ID (μmol/L)	TD (μmol/L)	ANOVA	Bonferroni Correction		
					ASD vs. TD	ASD vs. ID	TD vs. ID
Ala	238.8 ± 69.3	236.9 ± 73.7	269.41 ± 91.5	F = 1.522 *p* = 0.224 η^2^_p_ = 0.03	*p* = 0.354	*p* = 1.000	*p* = 0.45
Arg	22.25 ± 12.3	16.81 ± 7.6	18.75 ± 12.9	F = 1.751 *p* = 0.180 η^2^_p_ = 0.04	*p* = 0.678	*p* = 0.238	*p* = 1
Cit	22.52 ± 4.8	22.3 ± 4.9	21.8 ± 5.8	F = 0.177 *p* = 0.838 η^2^_p_ = 0.004	*p* = 1	*p* = 1	*p* = 1
Gln	458.2 ± 109.4	459.6 ± 97.9	450.5 ± 122.8	F = 0.052 *p* = 0.950 η^2^_p_ = 0.001	*p* = 1	*p* = 1	*p* = 1
Glu	86.8 ± 37.3	85 ± 26	87.7 ± 26.2	F = 0.045 *p* = 0.956 η^2^_p_ = 0.001	*p* = 1	*p* = 1	*p* = 1
Gly	226.3 ± 66.2	222.4 ± 50	247.2 ± 99.1	F = 0.861 *p* = 0.426 η^2^_p_ = 0.02	*p* = 0.787	*p* = 1	*p* = 0.75
Leu	125.2 ± 25.5	118.4 ± 22.1	131.4 ± 32.4	F = 1.395 *p* = 0.253 η^2^_p_ = 0.03	*p* = 1	*p* = 1	*p* = 0.253
Met	11.4 ± 4.4	11 ± 2.9	12.3 ± 3.5	F = 0.741 *p* = 0.480 η^2^_p_ = 0.02	*p* = 1	*p* = 1	*p* = 0.721
Orn	71.4 ± 17.3	78.2 ± 13.6	99.9 ± 61.2	F = 5.148 * *p* = 0.008 * η^2^_p_ = 0.11	*p* = 0.006 *	*p* = 1	*p* = 0.119
Phe	39.6 ± 7.8	40.5 ± 6.5	44.7 ± 10	F = 3.301 * *p* = 0.042 * η^2^_p_ = 0.07	*p* = 0.043 *	*p* = 1	*p* = 0.226
Pro	107.5 ± 37.9	116.6 ± 50	116.9 ± 46.6	F = 0.503 *p* = 0.607 η^2^_p_ = 0.01	*p* = 1	*p* = 1	*p* = 1
Tyr	47.9 ± 9.3	53.30 ± 11.6	58.7 ± 21.3	F = 4.560 * *p* = 0.013 * η^2^_p_ = 0.1	*p* = 0.010 *	*p* = 0.492	*p* = 0.581
Val	158.1 ± 34.1	54.4 ± 34.5	155.2 ± 35.4	F = 0.101 *p* = 0.904 η^2^_p_ = 0.002	*p* = 1	*p* = 1	*p* = 1

ASD = Autism Spectrum Disorder; ID = Intellectual Disability; TD = Typical Development; * = statistically significant; the η^2^_p_ can be interpreted as the percentage of the variance in the criterion variable which is explained by the grouping variable.

**Table 2 children-10-00412-t002:** Amino acid ratios comparison with ANOVA test in the three studied groups and paired post hoc comparisons with Bonferroni pairwise correction.

Amino Acid Ratio	ASD (μmol/L)	ID (μmol/L)	TD (μmol/L)	ANOVA	Bonferroni Correction		
					ASD vs. TD	ASD vs. ID	TD vs. ID
Leu+Val/Phe+Tyr	3.3 ± 0.5	2.9 ± 0.5	2.8 ± 0.5	F = 6.641 * *p* = 0.002 * η^2^_p_ = 0.13 *	*p* = 0.003 *	*p* = 0.053	*p* = 1
Cit/Arg	1.5 ± 1.3	1.7 ± 1.3	1.8 ± 1.3	F = 0.255 *p* = 0.776 η^2^_p_ = 0.006	*p* = 1	*p* = 1	*p* = 1
Cit/Phe	0.6 ± 0.1	0.6 ± 0.1	0.5 ± 0.2	F = 2.176 *p* = 0.120 η^2^_p_ = 0.05	*p* = 0.121	*p* = 1	*p* = 0.702
Leu/Phe	3.2 ± 0.5	3 ± 0.5	3 ± 0.6	F = 1.990 *p* = 0.143 η^2^_p_ = 0.04	*p* = 0.350	*p* = 0.267	*p* = 1
Met/Leu	0.1 ± 0.03	0.1 ± 0.03	0.1 ± 0.02	F = 0.348 *p* = 0.707 η^2^_p_ = 0.008	*p* = 1	*p* = 1	*p* = 1
Met/Phe	0.3 ± 0.1	0.3 ± 0.1	0.3 ± 0.1	F = 0.052 *p* = 0.950 η^2^_p_ = 0.001	*p* = 1	*p* = 1	*p* = 1
Phe/Tyr	0.8 ± 0.2	0.8 ± 0.2	0.9 ± 0.4	F = 0.435 *p* = 0.649 η^2^_p_ = 0.01	*p* = 1	*p* = 1	*p* = 1
Glu/Gln	0.2 ± 0.1	0.2 ± 0.1	0.2 ± 0.1	F = 0.138 *p* = 0.871 η^2^_p_ = 0.003	*p* = 1	*p* = 1	*p* = 1
Tyr/Leu	0.4 ± 0.1	0.5 ± 0.1	0.5 ± 0.1	F = 5.214 * *p* = 0.007 * η^2^_p_ = 0.11 *	*p* = 0.044 *	*p* = 0.017 *	*p* = 1
TyMet	5 ± 2.5	5.2 ± 1.8	4.9 ± 1.4	F = 0.090 *p* = 0.914 η^2^_p_ = 0.002	*p* = 1	*p* = 1	*p* = 1
Val/Phe	4+ 0.72	3.9 ± 0.7	3.6 ± 0.7	F = 3.738 * *p* = 0.028 ;η^2^_p_ = 0.08 *	*p* = 0.023 *	*p* = 0.927	*p* = 0.473

ASD = Autism Spectrum Disorder; ID = Intellectual Disability; TD = Typical Development; * = statistically significant; The η^2^_p_ can be interpreted as the percentage of the variance in the criterion variable which is explained by the grouping variable.

**Table 3 children-10-00412-t003:** ASD subjects (%) presenting with individual amino acid or ratio outside the reference range (RR).

Metabolite	TD Reference Range (10th and 90th Percentile) (μmol/L)	ASD below RR (μmol/L)	ASD above RR (μmol/L)
Ala	206.00–401.50	35%	0%
Arg	4.01–34.30	7.5%	17.5%
Cit	14.80–29.30	7.5%	2.5%
Gln	283.00–596.00	5%	12.5%
Glu	62.50–130.00	12.5%	7.5%
Gly	122.00–350.50	7.5%	7.5%
Leu	91.00–172.00	5%	5%
Met	8.61–16.45	22.5%	10%
Orn	56.70–141.50	15%	0%
Phe	33.40–60.55	17.5%	2.5%
Pro	59.40–164.50	5%	10%
Tyr	40.20–86.15	20%	0%
Val	117.00–210.50	7.5%	12.5%
Leu+Val/Phe+Tyr	2.17–3.41	0%	30%
Cit/Arg	0.69–2.90	17.5%	7.5%
Cit/Phe	0.35–0.69	2.5%	22.5%
Leu/Phe	2.58–3.88	7.5%	5%
Met/Leu	0.07–0.13	20%	7.5%
Met/Phe	0.18–0.37	17.5%	17.5%
Phe/Tyr	0.54–0.98	0%	15%
Glu/Gln	0.13–0-28	7.5%	12.5%
Tyr/Leu	0.32–0-58	10%	0%
Tyr/Met	3.34–6.94	20%	12.5%
Val/Phe	2.74–4.43	5%	35%

ASD = Autism Spectrum Disorder; ID = Intellectual Disability; TD = Typical Development.

**Table 4 children-10-00412-t004:** Spearman correlation among metabolites and clinical features of ASD group.

Metabolite	IQ/DQ	ADOS-2 SA	ADOS-2 RRB	ADOS-2 TS
Ala	r = 0.0034	r = −0.3080	r = −0.1343	r = −0.2980
	*p* = 0.9839	*p* = 0.0812	*p* = 0.4562	*p* = 0.0921
Arg	r = 0.0452	r = 0.0467	r = 0.2065	r = 0.1075
	*p* = 0.7904	*p* = 0.7965	*p* = 0.2490	*p* = 0.5514
Cit	r = 0.0196	r = 0.0825	r = 0.4801 *	r = 0.2287
	*p* = 0.9082	*p* = 0.6479	*p* = 0.0047 *	*p* = 0.2005
Gln	r = −0.0644	r = 0.2175	r = 0.0066	r = 0.0066
	*p* = 0.7048	*p* = 0.2241	*p* = 0.9708	*p* = 0.9708
Glu	r = −0.1589	r = 0.1285	r = −0.1926	r = 0.0420
	*p* = 0.3476	*p* = 0.4762	*p* = 0.2829	*p* = 0.0.8209
Gly	r = 0.1389	r = 0.0153	r = 0.0503	r = 0.0294
	*p* = 0.4124	*p* = 0.9327	*p* = 0.7810	*p* = 0.8709
Leu	r = 0.1676	r = 0.0500	r = −0.0645	r = 0.0195
	*p* = 0.3213	*p* = 0.7823	*p* = 0.7215	*p* = 0.9144
Met	r = −0.0063	r = −0.0574	r = 0.2053	r = 0.0209
	*p* = 0.9707	*p* = 0.7470	*p* = 0.2516	*p* = 0.9081
Orn	r = −0.01176	r = 0.0021	r = 0.2628	r = 0.0898
	*p* = 0.4881	*p* = 0.9908	*p* = 0.1395	*p* = 0.6192
Phe	r = −0.1359	r = −0.0420	r = 0.1144	r = 0.0038
	*p* = 0.4227	*p* = 0.8164	*p* = 0.5261	*p* = 0.9831
Pro	r = −0.0830	r = 0.0251	r = 0.1149	r = 0.0591
	*p* = 0.6251	*p* = 0.8896	*p* = 0.05244	*p* = 0.7437
Tyr	r = 0.1805	r = −0.0565	r = 0.1749	r = 0.0122
	*p* = 0.2850	*p* = 0.7548	*p* = 0.3303	*p* = 0.9463
Val	r = 0.0514	r = −0.1138	r = 0.0639	r = −0.0721
	*p* = 0.7627	*p* = 0.5283	*p* = 0.7238	*p* = 0.6902
Leu+Val/Phe+Tyr	r = 0.0694	r = 0.0050	r = −0.2094	r = −0.0661
	*p* = 0.6830	*p* = 0.9780	*p* = 0.2421	*p* = 0.7148
Cit/Arg	r = −0.1241	r = 0.0429	r = −0.1872	r = −0.0275
	*p* = 0.4643	*p* = 0.8124	*p* = 0.2969	*p* = 0.8794
Cit/Phe	r = 0.1019	r = 0.1080	r = 0.3341	r = 0.2007
	*p* = 0.5482	*p* = 0.5498	*p* = 0.0574	*p* = 0.2628
Leu/Phe	r = 0.2826	r = 0.1491	r = −0.1977	r = 0.0562
	*p* = 0.0901	*p* = 0.4077	*p* = 0.2701	*p* = 0.7651
Met/Leu	r = −0.1353	r = −0.0700	r = 0.2952	r = 0.0415
	*p* = 0.4245	*p* = 0.6988	*p* = 0.0953	*p* = 0.8188
Met/Phe	r = 0.0718	r = −0.0526	r = 0.2172	r = 0.0296
	*p* = 0.6727	*p* = 0.7711	*p* = 0.2246	*p* = 0.8703
Phe/Tyr	r = −0.2990	r = 0.0003	r = −0.1152	r = −0.0518
	*p* = 0.0722	*p* = 0.9987	*p* = 0.3885	*p* = 0.7748
Glu/Gln	r = −0.1186	r = 0.1717	r = −0.2950	r = 0.0421
	*p* = 0.4843	*p* = 0.3395	*p* = 0.0956	*p* = 0.8159
Tyr/Leu	r = 0.0171	r = −0.0933	r = 0.2267	r = −0.0006
	*p* = 0.9200	*p* = 0.6057	*p* = 0.2046	*p* = 0.9972
Tyr/Met	r = 0.0165	r = 0.0902	r = −0.1493	r = 0.0240
	*p* = 0.9226	*p* = 0.6178	*p* = −0.4070	*p* = 0.8944
Val/Phe	r = 0.1407	r = −0.0887	r = −0.0674	r = −0.0955
	*p* = 0.4061	*p* = 0.6234	*p* = 0.7092	*p* = 0.5971

ASD = Autism Spectrum Disorder; * = statistically significant.

## Data Availability

The raw data supporting the conclusions of this manuscript will be made available by the authors, without undue reservation, to any qualified researcher.

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
