# Peer review of "Neuroactive Amino Acid Profile in Autism Spectrum Disorder: Results from a Clinical Sample"

_children, 2023, doi:10.3390/children10020412_

Round 1

Reviewer 1 Report

dear authors,

thank you for the submission. I like the topic – some suggestions.

some important literatures are being missed e.g.: Lower levels of exogenous amino acids support their potential involvement in autism. Even if measuring exogenous amino acids in plasma cannot be utilized as a diagnostic tool, it can nevertheless help with the management of autistic individuals.

Bugajska J, Berska J, Wojtyto T, Bik-Multanowski M, Sztefko K. The amino acid profile in blood plasma of young boys with autism. Psychiatr Pol. 2017 Apr 30;51(2):359-368. English, Polish. doi: 10.12740/PP/65046. Epub 2017 Apr 30. PMID: 28581543.

Methods need participants recruitment flow diagram.

The raw data supporting the conclusions of this manuscript will be made available by the authors, without undue reservation, to any qualified researcher. Need to be publicly available.

Table 1 and Table 2 and Table 3 need effect size. Eta squared is a measure of effect size that is commonly used in ANOVA models.

Table 4 direction is not enough we need effect size.

The discussion needs to be rewritten and be much softer in the conclusion.

Author Response

Reviewer 1

Thank you very much for your revisions and comments. Hereby please you find a point-by-point reply to your revisions.

Dear authors,

thank you for the submission. I like the topic – some suggestions.

some important literatures are being missed e.g.: Lower levels of exogenous amino acids support their potential involvement in autism. Even if measuring exogenous amino acids in plasma cannot be utilized as a diagnostic tool, it can nevertheless help with the management of autistic individuals.

Bugajska J, Berska J, Wojtyto T, Bik-Multanowski M, Sztefko K. The amino acid profile in blood plasma of young boys with autism. Psychiatr Pol. 2017 Apr 30;51(2):359-368. English, Polish. doi: 10.12740/PP/65046. Epub 2017 Apr 30. PMID: 28581543.

Reply: We appreciated the Reviewer suggestion. A comment on the issue raised has been reported in the Introduction with the related reference by Bugajska et al (2017).

Methods need participants recruitment flow diagram.

Reply: We agree that participants' recruitment procedure maight be depicted in a flow diagram which now has been added to the text as Figure 1.

The raw data supporting the conclusions of this manuscript will be made available by the authors, without undue reservation, to any qualified researcher. Need to be publicly available.

Reply: Raw data and statistic analyses supporting the conclusions have been extensively reported in the revised text (see Tables 1-4) and are all absolutely  available upon request.

Table 1 and Table 2 and Table 3 need effect size. Eta squared is a measure of effect size that is commonly used in ANOVA models.

Reply: The data of Table 2 have been included in Table 1, since they represent the Post Hoc of the one-way ANOVA reported in Table 1. Previous Table 2 was then deleted. The effect size was calculated and reported for all the data of Tables 1 (amino acids) and 2 (now, amino acid ratios).

Table 4 direction is not enough we need effect size.

Reply: According with this and other reviewers table 4 was deleted because effect size measures were not available in the studies listed in previous table 4.  In the revised manuscript, an updated literature search supports data discussion including one meta-analysis on glutamate in ASD reporting effect size measures (Zheng et al. 2016).

The discussion needs to be rewritten and be much softer in the conclusion.

Reply: We considered this Reviewer's insight: the discussion was rewritten and the conclusions have been downsized.

Reviewer 2 Report

There is a plenty of studies reporting amino acid levels of ASD, however many studies have overlooked the impotence of comparing amino acid ratios between ASD and neurotypical subjects. The scientific merit of the present study lies in the novel technology (tandem mass spectrometry) used and examine the amino acids in terms of absolute concentrations as well as ratios. My main comments are mainly pertaining to the under-reporting of negative findings, not indicating the number of subjects with levels exceeding reference limits.  and the lack of critical analysis of the results with regard to amino acid metabolism.  Major comments Line 17; Line 32; ASD criteria include deficits in not only in social communication but also social interaction  There is a text revision available for DSM-5; I would suggest citing the latest. American Psychiatric Association. Diagnostic and Statistical Manual of Mental Disorders, Fifth Edition, Text Revision; American psychiatric association Washington, DC: 2022. Line 55-56; According to your statement “study included 89 patients admitted to the Child Neurology and Psychiatry Unit, Department of Clinical and Experimental Medicine”. These 89 subjects also include 27 subjects with typical development. What was the reason for admitting 27 neurotypical subjects? You have later on indicated that neurotypicals are “community controls”. isn’t it actually only 62 (89 - 27) patients that got admitted? Please resolve this confusion by revising. After going through 55-79 I feel I would have been clearer if you could organize the information into 3 paragraphs, indicating inclusion and exclusion criteria for each so that many of my comments can be resolved easily.1st paragraph; for ASD2nd paragraph; for intellectual disability (ID)3rd paragraph; for neurotypical Only the statistically significant amino acids and ratios are tabulated. This is under-reporting. Moreover, other amino acids and ratios will be useful for future systematic reviews/meta-analyses as both significant and non-significant effects are pooled. Therefore, include all other parameters in the same tables or separate tables as you wish. On the other hand, even though the mean difference is not significant, scientists are interested in the trend of the difference. Comparison of the mean will give a positive correlation only when an amino acid abnormality represents a major/common metabotype. Therefore I would suggest establishing reference ranges and determining how may ASD subjects have a specific amino acid /ratio outside the reference range. Consult the table 10 of the following paper. Adams JB, Audhya T, McDonough-Means S, Rubin RA, Quig D, Geis E, et al. Nutritional and metabolic status of children with autism vs. neurotypical children, and the association with autism severity. Nutr Metab (Lond). 2011;8(1):34.    Meta-analysis provides stronger evidence compared to individual research. The authors may consult the following systematic review/meta-analyses to compare their findings with pooled effect size. its good to cite them in discussion.    Zheng Z, Zhu T, Qu Y, Mu D. Blood Glutamate Levels in Autism Spectrum Disorder: A Systematic Review and Meta-Analysis. PLoS One. 2016;11(7):e0158688. Chen L, Shi XJ, Liu H, Mao X, Gui LN, Wang H, et al. Oxidative stress marker aberrations in children with autism spectrum disorder: a systematic review and meta-analysis of 87 studies (N = 9109). Transl Psychiatry. 2021;11(1):15.    Table 4 doesn’t contain much research on amino acid profiles. The tabulated references are not truly representative. You may find some other studies by referring to the above systematic reviews. All the statistically significant ratios in your study are interestingly reflecting the balance between branched-chain amino acids (leucine and valine) and aromatic amino acids (Phenylalanine and tyrosine). But this observation is not discussed. Authors should discuss this imbalance and try to relate it to the pathogenesis of ASD. the mutations of amino acid transporters and enzymes involved in the metabolism of branched-chain amino acids are relevant in ASD. For instance, Mutations in branched-chain ketoacid dehydrogenase kinase (BCKDK) are associated with autism spectrum disorder (Novarino et al., 2012, Du et al., 2022). This is not discussed.  

NOVARINO, G., EL-FISHAWY, P., KAYSERILI, H., MEGUID, N. A., SCOTT, E. M., SCHROTH, J., SILHAVY, J. L., KARA, M., KHALIL, R. O., BEN-OMRAN, T., ERCAN-SENCICEK, A. G., HASHISH, A. F., SANDERS, S. J., GUPTA, A. R., HASHEM, H. S., MATERN, D., GABRIEL, S., SWEETMAN, L., RAHIMI, Y., HARRIS, R. A., STATE, M. W. & GLEESON, J. G. 2012. Mutations in BCKD-kinase lead to a potentially treatable form of autism with epilepsy. Science, 338, 394-7.

DU, C., LIU, W. J., YANG, J., ZHAO, S. S. & LIU, H. X. 2022. The Role of Branched-Chain Amino Acids and Branched-Chain α-Keto Acid Dehydrogenase Kinase in Metabolic Disorders. Front Nutr, 9, 932670.

Minor comments MS denotes mass spectrometry and MS/MS denotes tandem mass spectrometry; therefore “Tandem MS/MS” is a redundant and technically wrong use of the terms. I would abbreviate it as “electrospray ionization-tandem mass spectrometry (ESI-MS/MS)”. Search the abbreviation “ESI-MS/MS” on the internet and you will get my point. But in the abstract, it’s better to avoid abbreviations (e.g. MS/MS) but use “electrospray ionization-tandem mass spectrometry”Use the abbreviations consistentlye.g. you have abbreviated “neurotypical developmental controls (TD)” at “Line 112”; But this abbreviation should have ideally been introduced at line 58 and used consistently afterward (e.g. even at line 99).    In tables 2 and 3, I think commas have been typed mistakenly instead of full stops.For instance, in Table 3, It should be 3.26, but not 3,26. isn’t it?TABLE 2,  71,39  or 71.39? In both tables, I have so many doubts about the numbers. Please check them all.           

Author Response

Reviewer 2

Thank you very much for your revisions and comments. Hereby please find a point-by-point reply to your revisions.

There is a plenty of studies reporting amino acid levels of ASD, however many studies have overlooked the impotence of comparing amino acid ratios between ASD and neurotypical subjects. The scientific merit of the present study lies in the novel technology (tandem mass spectrometry) used and examine the amino acids in terms of absolute concentrations as well as ratios.

Reply: thank you very much indeed for appreciating our efforts to face this topic.

My main comments are mainly pertaining to the under-reporting of negative findings, not indicating the number of subjects with levels exceeding reference limits and the lack of critical analysis of the results with regard to amino acid metabolism.  

Major comments Line 17; Line 32; ASD criteria include deficits in not only in social communication but also social interaction  There is a text revision available for DSM-5; I would suggest citing the latest. American Psychiatric Association. Diagnostic and Statistical Manual of Mental Disorders, Fifth Edition, Text Revision; American psychiatric association Washington, DC: 2022.

Reply: We absolutely agree with the Reviewer and we revised all text by reporting all raw data and statistical analyses for all measured metabolites (Tables 1-4). We updated the ref. to DSM-5 criteria as suggested.

 Line 55-56; According to your statement “study included 89 patients admitted to the Child Neurology and Psychiatry Unit, Department of Clinical and Experimental Medicine”. These 89 subjects also include 27 subjects with typical development. What was the reason for admitting 27 neurotypical subjects? You have later on indicated that neurotypicals are “community controls”. isn’t it actually only 62 (89 - 27) patients that got admitted? Please resolve this confusion by revising. After going through 55-79 I feel I would have been clearer if you could organize the information into 3 paragraphs, indicating inclusion and exclusion criteria for each so that many of my comments can be resolved easily.1st paragraph; for ASD2nd paragraph; for intellectual disability (ID)3rd paragraph; for neurotypical 

Reply: Thank you for the suggestions. To better report the recruitment process a flow diagram was shown in Figure 1. Following your suggestions, inclusion and exclusion criteria for enrollment were reported in the flow diagram and separately for the three groups.

Only the statistically significant amino acids and ratios are tabulated. This is under-reporting. Moreover, other amino acids and ratios will be useful for future systematic reviews/meta-analyses as both significant and non-significant effects are pooled. Therefore, include all other parameters in the same tables or separate tables as you wish. On the other hand, even though the mean difference is not significant, scientists are interested in the trend of the difference. 

Reply:  We revised all text by reporting all raw data and statistical analyses for all measured metabolites (Tables 1-4).

Comparison of the mean will give a positive correlation only when an amino acid abnormality represents a major/common metabotype. Therefore I would suggest establishing reference ranges and determining how may ASD subjects have a specific amino acid /ratio outside the reference range. Consult the table 10 of the following paper. Adams JB, Audhya T, McDonough-Means S, Rubin RA, Quig D, Geis E, et al. Nutritional and metabolic status of children with autism vs. neurotypical children, and the association with autism severity. Nutr Metab (Lond). 2011;8(1):34.   

Reply: Thank you for your valuable suggestions. We  established normal reference ranges in the TD group and assessed the tenth and ninetieth percentiles of their distribution. Then we evaluated the number of ASD subjects (%) presenting with individual amino acids or ratios outside the reference range (RR) (Table 3). These data are discussed in light of pertinent references.

Meta-analysis provides stronger evidence compared to individual research. The authors may consult the following systematic review/meta-analyses to compare their findings with pooled effect size. its good to cite them in the discussion.    Zheng Z, Zhu T, Qu Y, Mu D. Blood Glutamate Levels in Autism Spectrum Disorder: A Systematic Review and Meta-Analysis. PLoS One. 2016;11(7):e0158688. Chen L, Shi XJ, Liu H, Mao X, Gui LN, Wang H, et al. Oxidative stress marker aberrations in children with autism spectrum disorder: a systematic review and meta-analysis of 87 studies (N = 9109). Transl Psychiatry. 2021;11(1):15.    Table 4 doesn’t contain much research on amino acid profiles. The tabulated references are not truly representative. You may find some other studies by referring to the above systematic reviews. 

Reply: According to you and other reviewers, table 4 was deleted. In the revised manuscript, an updated literature search supports data discussion. The two systematic reviews you suggested were included and pertinent data from meta-analyses are reported in the discussion.

All the statistically significant ratios in your study are interestingly reflecting the balance between branched-chain amino acids (leucine and valine) and aromatic amino acids (Phenylalanine and tyrosine). But this observation is not discussed. Authors should discuss this imbalance and try to relate it to the pathogenesis of ASD. the mutations of amino acid transporters and enzymes involved in the metabolism of branched-chain amino acids are relevant in ASD. For instance, Mutations in branched-chain ketoacid dehydrogenase kinase (BCKDK) are associated with autism spectrum disorder (Novarino et al., 2012, Du et al., 2022). This is not discussed.  

NOVARINO, G., EL-FISHAWY, P., KAYSERILI, H., MEGUID, N. A., SCOTT, E. M., SCHROTH, J., SILHAVY, J. L., KARA, M., KHALIL, R. O., BEN-OMRAN, T., ERCAN-SENCICEK, A. G., HASHISH, A. F., SANDERS, S. J., GUPTA, A. R., HASHEM, H. S., MATERN, D., GABRIEL, S., SWEETMAN, L., RAHIMI, Y., HARRIS, R. A., STATE, M. W. & GLEESON, J. G. 2012. Mutations in BCKD-kinase lead to a potentially treatable form of autism with epilepsy. Science, 338, 394-7.

 DU, C., LIU, W. J., YANG, J., ZHAO, S. S. & LIU, H. X. 2022. The Role of Branched-Chain Amino Acids and Branched-Chain Keto Acid Dehydrogenase Kinase in Metabolic Disorders. Front Nutr, 9, 932670.

Reply: Thank you for your appreciated suggestions. We now report several insights on the observed amino acid unbalances in our study and we empowered the discussion also including some data from the above mentioned papers, all listed in the references.

Minor comments: MS denotes mass spectrometry and MS/MS denotes tandem mass spectrometry; therefore “Tandem MS/MS” is a redundant and technically wrong use of the terms. I would abbreviate it as “electrospray ionization-tandem mass spectrometry (ESI-MS/MS)”. Search the abbreviation “ESI-MS/MS” on the internet and you will get my point. 

Reply: this has been done as suggested.

But in the abstract, it’s better to avoid abbreviations (e.g. MS/MS) but use “electrospray ionization-tandem mass spectrometry” Use the abbreviations consistently e.g. you have abbreviated “neurotypical developmental controls (TD)” at “Line 112”; But this abbreviation should have ideally been introduced at line 58 and used consistently afterward (e.g. even at line 99).  

Reply: done as suggested.

In tables 2 and 3, I think commas have been typed mistakenly instead of full stops. For instance, in Table 3, It should be 3.26, but not 3,26. isn’t it?TABLE 2,  71,39  or 71.39? In both tables, I have so many doubts about the numbers. Please check them all.     

Reply:  this has been done as suggested.   

Reviewer 3 Report

Patients with ASD show abnormalities in the profile of certain plasma amino acids  (AA), including neuroactive amino acids. Monitoring plasma AA may be relevant for early diagnosis and intervention. The present cross-sectional study evaluated plasma AA profile in samples of 89 patients (40 subjects with ASD, males 32, age: 6.98 ± 4.03 years) and 22 with ID (males: 13,  age: 7.18 ± 2.97 years) by extracting dry blood spots by electrospray ionization-Tandem MS/MS system. 14 AA and 11 AA ratios were examined in patients with ASD, intellectual disability (ID) and neurotypical control subjects (TD). Clinical features, metabolic assessment, cognitive levels and measure of severity of autism symptoms were analyzed. Mean DQ/IQ measured in the ASD group was 64.94 ± 17.08 and in ID group 59.67 ± 14.54. Diagnosis of ASD was obtained according to DSM-5 criteria and ADI-R and ADOS (ASD group showed mean SA score of    14.24 ± 4.29 and mean RRB score of 3.48 ±1.75, indicating moderate to severe autism symptom severity). Children with Fragile X syndrome were excluded from the study. Among twenty-five metabolites analyzed in the three groups, six (24%) proved statistically significant differences: ornithine, phenylalanine, tyrosine, Leu+Val I Phe+Tyr, Tyr/Leu  and Val/Phe. The AA profile in the ASD group showed statistically lower levels of ornithine (p = 0.008), phenylalanine (p = 0.042) and tyrosine (p = 0.013). The statistically significant AA ratios were Leu+Val/Phe+Tyr (p = 0.002), Tyr/Leu (p = 0.007) and Val/Phe (p = 0.028), such differences remaining significant only in the comparison between ASD and TD. Finally, a positive correlation emerged between the score of the restricted and repetitive behavior on ADOS-2 and the citrulline levels in the ASD group (p = 0.0047). The authors conclude that patients with ASD may show a distinct   metabolic profile useful for studying their metabolic pathways in order to develop screening tests and targeted  therapies.

Major concerns.

The authors are commended for their effort to fill in a gap in the filed of ASD on this complex issue.

 As they reported in Introduction, literature on the AA profile in ASD has been variable, inconsistent, confusing, difficult to interpret. And despite overall good effort, their findings while commendable are also variable and hard to interpret.

The authors should continue to work on figuring out how to interpret and present the relevance and connect these AA variabilities to ASD phenotype, for example. That would be some meaningful contribution that others could replicate in amore meaningful way.  For example, among 40 subjects with ASD of that young age, examine level of the ASD severity, categorically, divide them into level 1 ASD (mild) vs severe level 2/3 (Kanner type). And then apply the AA profiles in these two categories and contrast what is increased and decreased. You may do the same with ADOS scores that accompany these same subjects (ADOS, and ADI-R scores supplement the DSM-5). The authors can then adjust those findings for IQ, gender. And compare with smaller ID and TD groups to get some idea as the power of those two latter subgroups will also limit the interpretation.

I think that would give a more meaningful convolution of the AA data, to connect with ASD phenotype severity.

In the same line, this reviewer feels that the present study data and presented research on literature displayed

should be better tied, in your tables, graphs, to make clear point. Summarize valuable Tab 4 to support your findings, rather than to appear like a part of review paper.

The conclusion also has to be toned down as is. The AA findings are difficult to interpret in terms of what is meaningful about them, given the heterogenous ASD sample. “That patients with ASD may show a distinct metabolic profile, demonstrating that  this may be used to identify a subset of ASD patients with respect to TD and to possibly evaluate the severity of this condition” statement from the authors does not reflect the presented data but maybe achieved after the additional aforementioned analyses are completed.

Minor suggestions

Introduction

Line 34, ..”Intellectual disability occurs in 45% of ASD patients while  almost 30% have a regression or loss of acquired skills.”

Needs reference(s).

Lines 38-43

Needs references.

Line 55, “admitted to the Child Neuralogy and Psychiatry Units, Department of Clinical and Experimental Medicine, University of Catania, in a three-year period, between 2019 and 2021.”

the Child Neurology

Tables 1 and 2 and 4.. line 175, omithine values

Omithine

Should be ornithine.

Line 175 and throughout, …”autistic children,” should be ..children with ASD.’

Table 4 is useful but needs work as the above suggested.

Author Response

Reviewer 3

Thank you very much for your revisions and comments. Hereby please you find a point-by-point reply to your revisions.

Patients with ASD show abnormalities in the profile of certain plasma amino acids  (AA), including neuroactive amino acids. Monitoring plasma AA may be relevant for early diagnosis and intervention. The present cross-sectional study evaluated plasma AA profile in samples of 89 patients (40 subjects with ASD, males 32, age: 6.98 ± 4.03 years) and 22 with ID (males: 13,  age: 7.18 ± 2.97 years) by extracting dry blood spots by electrospray ionization-Tandem MS/MS system. 14 AA and 11 AA ratios were examined in patients with ASD, intellectual disability (ID) and neurotypical control subjects (TD). Clinical features, metabolic assessment, cognitive levels and measure of severity of autism symptoms were analyzed. Mean DQ/IQ measured in the ASD group was 64.94 ± 17.08 and in ID group 59.67 ± 14.54. Diagnosis of ASD was obtained according to DSM-5 criteria and ADI-R and ADOS (ASD group showed mean SA score of    14.24 ± 4.29 and mean RRB score of 3.48 ±1.75, indicating moderate to severe autism symptom severity). Children with Fragile X syndrome were excluded from the study. Among twenty-five metabolites analyzed in the three groups, six (24%) proved statistically significant differences: ornithine, phenylalanine, tyrosine, Leu+Val Phe+Tyr, Tyr/Leu  and Val/Phe. The AA profile in the ASD group showed statistically lower levels of ornithine (p = 0.008), phenylalanine (p = 0.042) and tyrosine (p = 0.013). The statistically significant AA ratios were Leu+Val/Phe+Tyr (p = 0.002), Tyr/Leu (p = 0.007) and Val/Phe (p = 0.028), such differences remaining significant only in the comparison between ASD and TD. Finally, a positive correlation emerged between the score of the restricted and repetitive behavior on ADOS-2 and the citrulline levels in the ASD group (p = 0.0047). The authors conclude that patients with ASD may show a distinct   metabolic profile useful for studying their metabolic pathways in order to develop screening tests and targeted  therapies.

Major concerns.

The authors are commended for their effort to fill in a gap in the filed of ASD on this complex issue.

 As they reported in Introduction, literature on the AA profile in ASD has been variable, inconsistent, confusing, difficult to interpret. And despite overall good effort, their findings while commendable are also variable and hard to interpret.

Reply: We thank the reviewer for positive comments addressing the difficulty to interpret amino acid unbalances in children with ASD.

The authors should continue to work on figuring out how to interpret and present the relevance and connect these AA variabilities to ASD phenotype, for example. That would be some meaningful contribution that others could replicate in a more meaningful way.  For example, among 40 subjects with ASD of that young age, examine level of the ASD severity, categorically, divide them into level 1 ASD (mild) vs severe level 2/3 (Kanner type). And then apply the AA profiles in these two categories and contrast what is increased and decreased. You may do the same with ADOS scores that accompany these same subjects (ADOS, and ADI-R scores supplement the DSM-5). The authors can then adjust those findings for IQ, gender. And compare with smaller ID and TD groups to get some idea as the power of those two latter subgroups will also limit the interpretation. I think that would give a more meaningful convolution of the AA data, to connect with ASD phenotype severity.

Reply: We appreciate the reviewer's comments. For the sake of clarity we now report in the revised text all the correlations we measured among all studied metabolites and clinical variables (Cognitive level (IQ/DQ), Autism severity: ADOS-Social Affect score, ADOS Restricted/Repetitive Behavior score, and ADOS Total Score). We understand that reporting a new data set comparing possible differences in the amino acid profiles of different clusters of patients could be the best. However, we realized that this is not affordable with the present clinical sample. Thus we detailed such crucial points among study limitations as follows:”It is important to take into account the intrinsic heterogeneity of symptoms and severity in ASD. In particular, the large majority of studied patients with ASD were cognitively impaired and had moderate to severe autism symptom severity. Such factors prevented us to compare the amino acid profiles between ASD subjects with different severity levels, according with the DSM-5 definition (level 1 vs levels 2-3). Moreover, the recruited participants were rather homogeneous in ages precluding us to compare younger and older participants with regard to blood amino acid variations”.

In the same line, this reviewer feels that the present study data and presented research on literature displayed should be better tied, in your tables, graphs, to make clear point. Summarize valuable Tab 4 to support your findings, rather than to appear like a part of review paper.

Reply: Thank you for your appreciated comments. According to you and other reviewers, table 4 was deleted. In the revised manuscript, an updated literature search supports data discussion. Two systematic reviews were included in the references and pertinent data from meta-analyses are reported in the discussion. We now report several insights on the observed amino acid unbalance in our study and we do hope this might empower the discussion.

The conclusion also has to be toned down as is. The AA findings are difficult to interpret in terms of what is meaningful about them, given the heterogenous ASD sample.

“That patients with ASD may show a distinct metabolic profile, demonstrating that this may be used to identify a subset of ASD patients with respect to TD and to possibly evaluate the severity of this condition” statement from the authors does not reflect the presented data but maybe achieved after the additional aforementioned analyses are completed.

Reply: We agree with the Reviewer's opinion. The conclusions have been downsized and any reference to diagnostic power and severity of conditions has been removed. Based on present findings and pertinent data from the medical literature, we underline that albeit with existing controversial results, present and literature findings support amino acid profile analyses in ASD patients to better understanding those metabolic pathways amenable of therapeutic interventions.   

Minor suggestions

Introduction

Line 34, ..”Intellectual disability occurs in 45% of ASD patients while  almost 30% have a regression or loss of acquired skills.”

Needs reference(s).

Lines 38-43

Needs references.

Reply: References have been added

Line 55, “admitted to the Child Neuralogy and Psychiatry Units, Department of Clinical and Experimental Medicine, University of Catania, in a three-year period, between 2019 and 2021.”

the Child Neurology

Reply: Child Neurology 

Tables 1 and 2 and 4.. line 175, omithine values

Omithine

Should be ornithine.

Reply: this has been done

Line 175 and throughout, …”autistic children,” should be ..children with ASD.’

Reply: sorry for the typos. This has been rewritten correctly.

Table 4 is useful but needs work as the above suggested.

Reply: as detailed in the previous paragraphs, previous table 4 was deleted.

Round 2

Reviewer 2 Report

Dear authors,

Thanks for fixing all the issues i pointed out. The paper reads well. 

I think its good to boldface the statistically significant values (e.g. values <0.05) in the tables. 

Reviewer 3 Report

The authors have addressed all concerns raised in a satisfactory or in an acceptable manner.